# On the Multi-Functional Behavior of Graphene-Based Nano-Reinforced Polymers

**DOI:** 10.3390/ma14195828

**Published:** 2021-10-05

**Authors:** Konstantina Zafeiropoulou, Christina Kostagiannakopoulou, Anna Geitona, Xenia Tsilimigkra, George Sotiriadis, Vassilis Kostopoulos

**Affiliations:** 1Department of Mechanical Engineering & Aeronautics, University Campus Patras, GR-26504 Rio, Achaia, Greece; k_zafeiropo@upnet.gr (K.Z.); kostagia@mech.upatras.gr (C.K.); ageitona@upatras.gr (A.G.); xenitsilimigkra@gmail.com (X.T.); sotiriad@mech.upatras.gr (G.S.); 2Foundation of Research and Technology, Institute of Chemical Engineering Sciences (FORTH/ICE-HT), Stadiou Str., GR-26504 Rio, Achaia, Greece

**Keywords:** epoxy, graphene nano-platelets (GNPs), multi-functional materials

## Abstract

The objective of the present study is the assessment of the impact performance and the concluded thermal conductivity of epoxy resin reinforced by layered Graphene Nano-Platelets (GNPs). The two types of used GNPs have different average thicknesses, <4 nm for Type 1 and 9–12 nm for Type 2. Graphene-based polymers containing different GNP loading contents (0.5, 1, 5, 10, 15 wt.%) were developed by using the three-roll mill technique. Thermo-mechanical (Tg), impact tests and thermal conductivity measurements were performed to evaluate the effect of GNPs content and type on the final properties of nano-reinforced polymers. According to the results, thinner GNPs were proven to be more promising in all studied properties when compared to thicker GNPs of the same weight content. More specifically, the glass transition temperature of nano-reinforced polymers remained almost unaffected by the GNPs inclusion. Regarding the impact tests, it was found that the impact resistance of the doped materials increased up to 50% when 0.5 wt.% Type 1 GNPs were incorporated within the polymer. Finally, the thermal conductivity of doped polymers with 15 wt.% GNPs showed a 130% enhancement over the reference material.

## 1. Introduction

In recent years, the use of nanomaterials as fillers in the production of nano-reinforced polymers, has attracted significant interest due to their unique properties. The key factor for the increasing demand of the polymer nanocomposites is their multi-functionality as a result of the nano-metric additives that drastically enhance their performance.

In line with this direction, most of the research has focused on polymer nanocomposites based on carbon nanospecies (CNSs). Numerous carbon additives, such as carbon black (CB), carbon nanotubes (CNTs) and carbon nanofibers (CNFs), have been utilized to enhance the properties of pure polymers [1,2,3,4,5]. Since their discovery, graphene nanospecies (GNSs) have demonstrated intriguing properties, including a high thermal and electrical conductivity, increased thermo-mechanical performance and superior mechanical strength comparable to the aforementioned carbon nanospecies [6,7,8,9,10,11,12,13].

Among the various forms of graphene, graphene nanoplatelets (GNPs) are the most widely used nanoparticles. GNPs are disk-shaped graphite particles, which comprise two or more graphene layers, resulting in a total thickness of 0.7 to 100 nm.

GNPs have gained more and more ground in high-performance applications due to their exceptional properties in multiple fields [14,15,16]. One of the major advantages of GNPs is that their incorporation in the polymer significantly enhances the thermal conductivity of the nanocomposites. More precisely, Kalaitzidou et al. [17] cited that polypropylene nanocomposites, reinforced with 25%vol. xGNPs, showed a 500% increase in thermal conductivity in comparison with the reference material. Furthermore, Yu et al. [18] presented an increment in thermal conductivity of more than 3000% by the addition of 25%vol. GNPs within the matrix of epoxy nanocomposites. Fang et al. [19] reported a significant improvement in the thermal conductivity of polystyrene films filled with 2 wt.% polystyrene-grafted graphene from 0.158 Wm^−1^K^−1^ to 0.413 Wm^−1^K^−1^. Moreover, Wang et al. [20] observed that the thermal conductivity of an epoxy composite loaded with 5 wt.% GNPs increased significantly up to 115% as compared to the pristine material. Compounds with different GNP contents (0–8 wt.%) were prepared by Wang et al. [21] to study the thermal conductivity of the nanocomposite. The content of 8 wt.% was proven to be more promising, reaching a value of 1.181 W/m K, which was increased by 627% compared to the neat epoxy. The structure of GNPs provide a 2D path, which contributes to a more effective phonon transport [22,23] and constitutes them as potential additives for applications requiring high thermal conductivity properties.

The thermo-mechanical behavior of graphene-based polymer composites has also been studied by researchers in relation to the effect of doped epoxy nanocomposites at the glass transition temperature (Tg). The study of Wang et al. [21] concluded that the Tg of the doped epoxy was enhanced increasing the GNPs concentration, regardless of the GNP particle size. More precisely, a 5 wt.% GNPs content achieved an increase in Tg of up to 4 °C in comparison with the pure epoxy material. Similar results were obtained by Yasmin et al. [24], who observed that pure epoxy Tg increased slightly with the incorporation of graphite platelets from 143 °C to 145 and 146 °C for graphite contents of 2.5 and 5 wt.%, respectively. It is conceivable that the presence of the nano-filler restricts the macromolecular movement in the nano-reinforced polymers, affecting the Tg of the polymer positively. The Tg value of nanocomposites shifts to a higher temperature, from 93.4 °C (pure epoxy) to 99.1 °C (graphene concentration 0.3 wt.%), according to the study of Wei et al. [25]. For concentrations above 0.5%wt, a drop in Tg was recorded due to the presence of large aggregates.

Many researchers are devoting their efforts to producing composites using GNPs/graphene as a novel filler for polymer matrices due to its outstanding mechanical properties. Kalaitzidou et al. [17] studied the flexural strength and modulus of various polypropylene composites modified with xGNPs, up to a loading level of 20%vol. They reported that the addition of 5%vol. xGNPs into the matrix increased the flexural strength of the final product by ~36%. Moreover, they also monitored an increase of 30% in the flexural modulus by the inclusion of 1%vol. xGNPs. In their study, Yasmin et al. [24] summarized that the epoxy nanocomposite reinforced with 2.5 wt.% and 5 wt.% graphite platelets showed an increase of about 10% and 25% in the elastic modulus, respectively, over the pure epoxy matrix. However, the variation in the tensile strength with the graphite concentration follows a reverse trend, resulting in an increase of the tensile strength of about 21% and 9% with the addition of 2.5 wt.% and 5 wt.%, respectively. The tensile and flexural strength of GNP nanocomposites with 0.3 wt.% loading content was reported by Wei et al. [25] and showed an increase of up to 12.6% and 10%, respectively. However, in both cases the maximum value of the modulus was reached at 1 wt.% of the graphene content. Using 0.125 wt.% functionalized graphene sheets, Rafiee et al. [26] presented the Young’s modulus of an epoxy nanocomposite to be 50% greater than the baseline epoxy. Similarly, the ultimate tensile strength measurements were observed to be increased by 45% for the respective content. Generally, it is claimed that the improvement of the tensile and flexural performance can be attributed to the fact that GNSs appear to have a high rigidity and excellent mechanical properties.

Fracture toughness (K_IC_) is also a crucial property for polymers and has been studied by several researchers. Zhang et al. [27] investigated the effect of different GNPs contents within a polymer in terms of its mode I fracture toughness (K_IC_). They claimed that K_IC_ increased by 50% when using 0.3 wt.% GNPs. After that content, K_IC_ decreased compared to the neat material. In the same direction, by using a GNPs content of 0.5 wt.%, an increase of 43% in K_IC_ was reported by Chandrasekaran et al. [28] and an increase of 76% was reported by Feng et al. [29]. A further increase of the GNPs content leads to a decrease of K_IC_ in both cases. At this point, a contradiction has been identified, since Chatterjee et al. [30] reported a continuous improvement of K_IC_ (up to 80%) with an increase of the GNPs content up to 2 wt.% Chandrasekaran et al. [31] also investigated the effect of the GNPs content on K_IC_, and they concluded that at 1 wt.% of the GNPs content they received the highest fracture toughness (a 49% increase compared to the reference epoxy system).

Limited work has been carried out on the Charpy impact energy absorption of polymers enhanced with GNPs. Shraddha et al. [32] observed a slight improvement of up to 10% when introducing 2 wt.% GNPs within an epoxy resin.

Therefore, it is obvious that there is a lack of consensus regarding the weight concentration of GNPs for the optimum improvement of the mechanical properties of epoxy nanocomposites. Furthermore, there are additional influential parameters such as the dispersion method/quality, the geometrical characteristics of GNPs (mean diameter, aspect ratio, specific surface area), and their possible pretreatment/functionalization that must be taken into serious consideration for the development of multi-functional nanocomposites.

The present study focuses on the development of graphene-based nano-reinforced polymers with improved fracture (Charpy impact) and thermal conductivity characteristics. An extensive investigation is carried out concerning the effect of the GNPs loading content on the above-mentioned properties that also includes their effect on Tg. Furthermore, the comparative results demonstrate that the aspect ratio of GNPs is possibly a crucial parameter for the final properties of nanocomposites.

## 2. Experimental Section

### 2.1. Materials

An epoxy B-stage system, obtained from Huntsman Advanced materials (Basel, Switzerland), was used in this study as the matrix material. This system contains four components: the low-viscosity epoxy resin Araldite LY1556, the hardener paste Aradur 1571, the accelerator paste 1573 and the polyamine hardener Aradur XB 3403, while their mixing ratio is 100:23:5:12 by weight.

The two types of graphene nanoplatelets (GNPs) used were 97% pure and supplied by Cheap Tube Inc. (Grafton, MA, USA). Type 1 GNPs have an average thickness of <4 nm, while Type 2 have an average thickness of 9–12 nm. Both types have the same lateral dimension of 2 microns, while the surface area is >750 m^2^/g for Type 1 and in the range of 600–750 m^2^/g for Type 2. The aspect ratio, which is defined as the ratio of the lateral dimension to the average thickness of GNPs, is calculated to be >500 for Type 1 and 170–220 for Type 2.

### 2.2. Preparation of Samples

Nano-modified epoxy mixtures were produced using a three roll-mill, also known as calender, whose process is described in detail by Kostagiannakopoulou et al. [33]. After mixing the appropriate amount of GNPs and epoxy resin inside a glove box for safety reasons, the dispersed mixture was fed through the feeder roll and was collected at the apron roll of the calender, repeating this five times for each gap setting in order to obtain the homogeneous graphene-based blends. Following this, the other three components of the B-stage system were added to the prepared suspension and degassed in a vacuum chamber to avoid air inclusion. Then, the developed material was poured into silicon rubber molds and was cured in an autoclave oven, according to the manufacturer’s instructions, 2 h at 120 °C and 6 bars pressure. Following the aforementioned process, graphene-based nano-reinforced polymers at different loading levels were developed: 0.5, 1, 5, 10 and 15 wt.% GNPs. Neat epoxy samples were also manufactured for reference. Figure 1 presents the manufacturing process in detail.

### 2.3. Testing Campaign

The glass transition temperature of the materials was calculated using the DMTA 983 of Du Pont (TA Instruments, Inc., New Castle, DE, USA). The specimens were subjected to Dynamic Mechanical Analysis (Figure 2a) tests at a frequency of 1 Hz, a heating rate of 2 °C/min and a temperature range of 25–250 °C. Three samples were used for each produced material, and the dimensions of the specimen were 50 mm × 10 mm × 3 mm.

The impact resistance was measured following the Charpy impact method by using a Karl Frank GMBH machine (Figure 2b). The measurements were calculated by ASTM D 6110 principles using V-notched specimens of 124.5 mm × 10.2 mm × 12.7 mm.

A TCi Mathis Analyzer (Figure 2c) was used for the evaluation of the thermal conductivity of the developed materials, providing a detailed overview of their thermal characteristics. Four specimens were measured for each tested material type. The dimensions of the measured samples were 25 mm × 25 mm × 5 mm. In both test campaigns, five samples of each loading content were used.

Finally, a rough estimation of the quality of dispersion of the nanofillers into the matrix was performed using Scanning Electron Microscopy (LEO SUPRA 35VP).

## 3. Results and Discussion

### 3.1. Thermo-mechanical Tests

Thermo-mechanical tests were performed to define the glass transition temperature of produced nano-reinforced polymers. The glass transition temperature (Tg) of the developed nanocomposites was calculated by DMA tests and presented in Figure 3. It is observed that the Tg values of nano-reinforced polymers are slightly lower compared to those of the reference material. The presence of GNPs affects the crosslinking density of epoxy by restricting the size of polymer chains, thus facilitating their movement, although the presence of GNPs acts as mobility obstacles. The decrease in crosslinking lowers the heat release rate, and though the Tg is observed to be lowered in the case of nano-modified epoxy compared to the neat material [27]. Thus, the addition of GNPs into the polymer results in a slight degradation of Tg. This fact is in contrast to what was expected based on the literature. Generally, it is known that the presence of nanoparticles restricts the macromolecular movement in the nano-reinforced polymer, positively affecting the Tg of the polymer. The nano-size of GNPs limits the partial motion of polymer chains and results in higher temperatures, transiting the material from a glassy to a rubbery state [34]. However, as is obvious, the results of the present study were not in agreement with this approach. According to the literature [24], the size of the polymer chains, the good adhesion between the filler and matrix, the geometry and weight concentration of nanofillers, and finally the curing conditions are basic parameters that can significantly affect Tg. Taking into consideration that the curing conditions were the same for all the tested materials, it is concluded that the presence of GNPs, the quality of the adhesion and the geometry of the fillers affect the size and the mobility of polymer chains and finally result in the slight decrease of Tg. It is also shown that the Type 2 GNPs affect the Tg more negatively than the Type 1 GNPs do.

### 3.2. Impact Tests

Impact experiments (Charpy tests) were conducted to study the contribution of Type 1 and Type 2 GNP materials to the impact resistance of the resin. Figure 4 depicts the energy absorption during the impact tests of the material per square meter. It is evident that the use of both GNP types proved efficient and led to a remarkable increase of the impact resistance compared to the neat material. Additionally, it was observed that the Type 1 material performed better at a content of 0.5 wt.% than Type 2 did, noting the maximum increase (50%) compared to the reference epoxy. The highest aspect ratio of Type 1 graphene promotes its improved impregnation by the polymer and thereby increases the contact surface between the two constituents (matrix and nano-reinforcement). This leads to the formation of a more extensive and stronger interface between the nanoparticles and the epoxy matrix, which results in a more efficient load transfer and the activation of additional energy absorbing mechanisms (pinning, bifurcation) activated by GNPs during fracture [31].

Additionally, it should be mentioned that the Type 1 presence in the polymer exhibited higher values of energy absorption compared to Type 2 GNPs at the same weight content. The explanation for this is that Type 1 graphene nanoplatelets, apart from having a higher surface area to that of Type 2 graphene, also have a higher aspect ratio and higher interlayer fracture resistance due to the limited number of graphene layers. Thus, they resist more to interlayers sliding and absorb more energy during impact damage. Furthermore, the Type 2 graphene nanoplatelets, due to their thicker structure, act as stress concentration sites and promote fracture initiation after impact.

It is obvious in Figure 4 that the GNPs content of 0.5 wt.%, of either Type 1 or Type 2, maximizes the impact resistance of nanocomposites. As the content increases for both types of graphene, the impact resistance decreases due to the formation of agglomerations within the polymer, which act as inherent imperfections. Specifically, a GNPs loading level higher than 5 wt.% has a detrimental effect on the impact resistance of nanocomposites.

### 3.3. Thermal Conductivity Measurements

The bar chart in Figure 5 represents the results of the thermal conductivity (TC) of the developed neat and nano-reinforced polymers. It is conceivable that the reference material appeared to have an almost insulating behavior, while the addition of GNPs into the polymer proved beneficial for the enhancement of the thermal conductivity of the final nanocomposites. The weight content of the GNPs and their aspect ratio are two key parameters explaining this behavior.

In particular, it is observed that the thermal conductivity of the resin system was significantly increased in the case of polymers that were reinforced with higher contents of GNPs (>5 wt.%) for both types. The explanation for this behavior is that with the increase of the GNPs content, the distance between the adjacent platelets is reduced, as is evident in the SEM images of Figure 6. This set-up leads to the reduction of the thin, insulated polymer film, which exists between the two constituents (matrix and filler), thus facilitating the phonon transport between them and consequently leading to a higher thermal conductivity. For a GNPs content higher than 5 wt.%, it is obvious that the thermal conductivity of the nano-reinforced polymer starts to increase when increasing the GNPs content in the polymer. Figure 5 shows that the highest increase of 130% occurred at the highest GNPs content (15 wt.%) οf Type 1 (thinner) GNPs in the polymer.

Moreover, polymers doped with Type 1 GNPs proved more conductive compared to Type 2 at the same weight content, for all loading levels (1, 5, 10 and 15 wt.%). The greatest difference between the thermal conductivity values was observed at the content of 15 wt.%, where polymers with a Type 1 reinforcement exhibited a 72% higher thermal conductivity than those with a Type 2 reinforcement. The thermal contact resistance between GNPs and the interfacial thermal resistance between the matrix and the GNPs considerably affect the thermal conductivity of the resulting nanocomposite. It is argued that the thermal contact resistance of polymers with Type 2 is much higher than that of Type 1, since the number of graphite interfaces of the conductive network is much higher and the phonon scattering will restrict the thermal conductivity.

Furthermore, it is believed that although there is a significant presence of agglomerates at high loading levels of GNPs, the higher aspect ratio of Type 1 GNPs promotes a better contact between the nanofillers and reduces the phonon scattering. The presence of the conductive channels, due to the direct contact of the GNPs, supports the better phonon transfer and therefore the heat transfer within the material. This is also facilitated by the lower number of layers of Type 1 GNPs, which also reduces the phonon scattering.

Table 1 summarizes the results concluded for the tested materials in all cases.

### 3.4. SEM

The cited SEM micrographs of the nano-reinforced polymers enhanced with different weight contents of Type 2 GNPs are illustrated in Figure 6. It can be observed from these SEM images that the GNPs are satisfactorily dispersed in the nano-reinforced polymers. However, it is clearly shown that increasing the filler concentration in the epoxy resin leads to the formation of a significant number of agglomerates between the GNPs, which are indicated by the white circles depicted in Figure 6b–f. Furthermore, in Figure 7 the agglomerations between the nanoparticles, the buckling and the rolling up of Type 2 GNPs are clearly observed in the case of the 0.5 wt.% GNP nano-reinforced polymer.

## 4. Conclusions

Epoxy polymer nanocomposites were manufactured and tested to investigate their Charpy impact and thermal conductivity properties and were compared to the reference material. GNPs at different weight contents (0.5, 1, 5, 10 and 15%) were incorporated within the epoxy system, and discrepancies were observed regarding the measured values. The GNP integration did not practically influence the glass transition temperature of nano-reinforced polymers. Regarding the Charpy impact properties, it was shown that the impact resistance was remarkably enhanced by up to 50% in the case where Type 1 GNPs at a 0.5 wt.% content were dispersed into the polymers. In addition, the thermal conductivity of nano-reinforced epoxy polymers increased with the inclusion of high GNPs contents (>5 wt.%) into the polymer. Specifically, the addition of 15 wt.% of Type 1 GNPs into the epoxy matrix caused a significant increase of 130% in the thermal conductivity of the developed materials compared to neat epoxy. It is clear that by increasing the weight content of GNPs in the epoxy polymer, the impact properties of the samples were degrading while the thermal conductivity was being enhanced. The dominance of Type 1 GNPs is obvious in improving both properties and is probably due to the different aspect ratios of the two types of GNPs. The glass transition temperature did not undergo any knock-down effect, particularly in the case of introducing Type 1 GNPs in the polymer, which resulted in an operational material. Based on the above results, the incorporation of GNPs endows the polymers with multi-functional properties and shows broad prospects for industrial applications. Thus, their integration in electronic, high-temperature dielectric and energy storage devices can be achieved while maintaining the thermo-mechanical performance at high levels.

## Figures and Tables

**Figure 1 materials-14-05828-f001:**
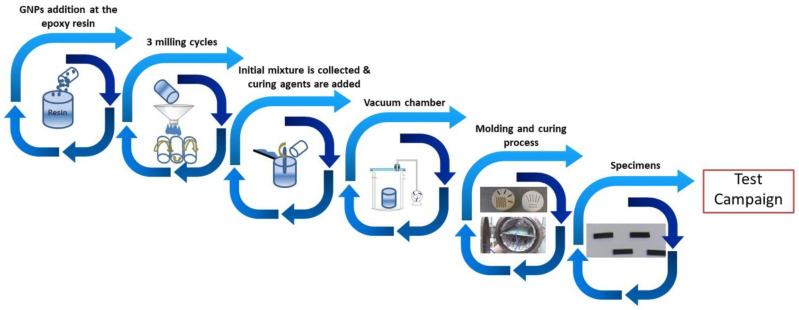
Manufacturing process of the tested specimens.

**Figure 2 materials-14-05828-f002:**
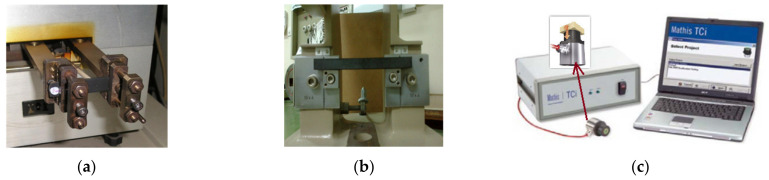
Experimental tools. Polymer specimen in the (**a**) DMA equipment, (**b**) Charpy Impact equipment, and (**c**) Thermal Conductivity set-up.

**Figure 3 materials-14-05828-f003:**
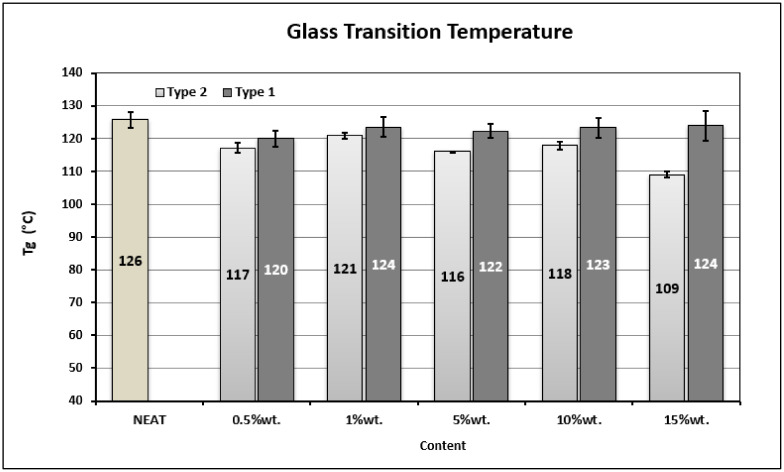
Thermo-mechanical properties of the produced polymers.

**Figure 4 materials-14-05828-f004:**
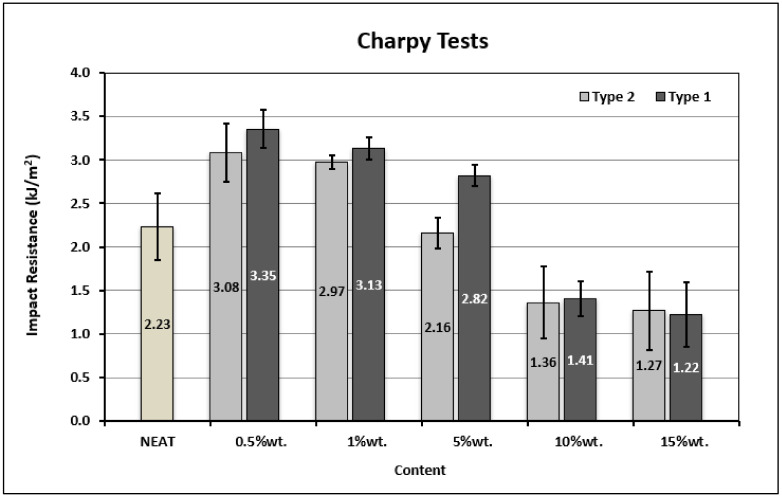
Impact properties of the nano-reinforced polymers.

**Figure 5 materials-14-05828-f005:**
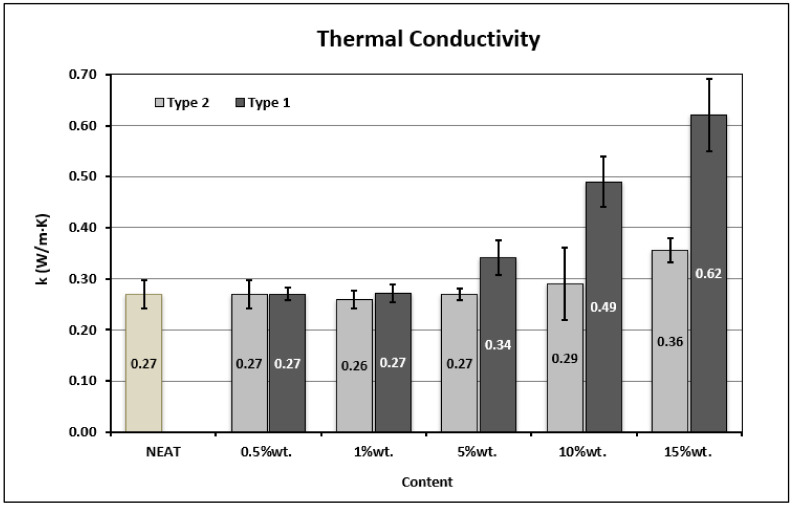
Thermal conductivity of the developed polymers.

**Figure 6 materials-14-05828-f006:**
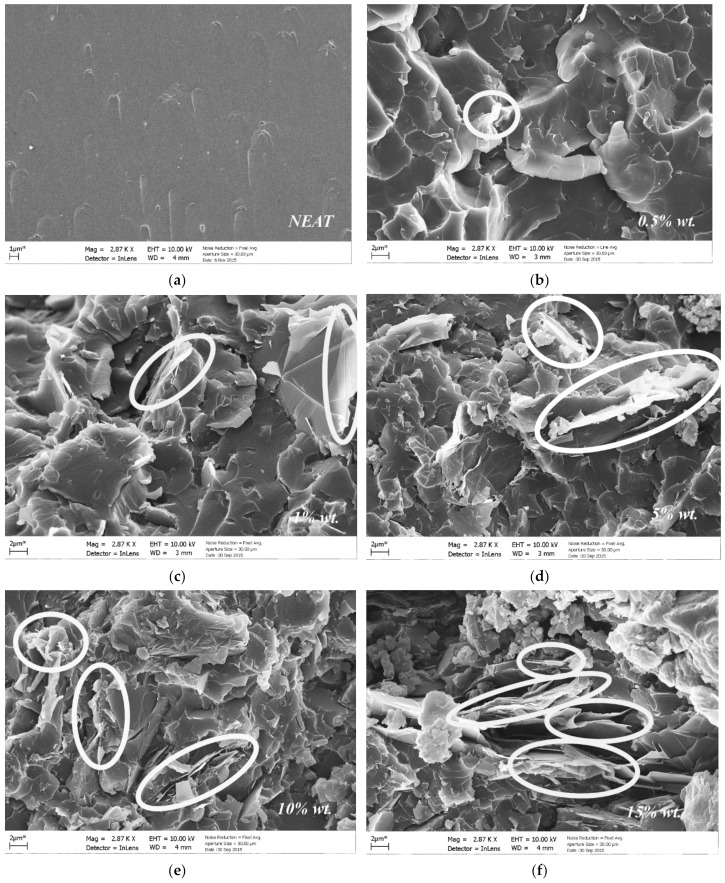
SEM images of the neat and nano-reinforced polymers enhanced with different contents of Type 2 GNPs: (**a**) neat, (**b**) 0.5 wt.%, (**c**) 1 wt.%, (**d**) 5 wt.%, (**e**) 10 wt.% and (**f**) 15 wt.% White circles indicate the agglomerates formed between the GNPs inside the polymers.

**Figure 7 materials-14-05828-f007:**
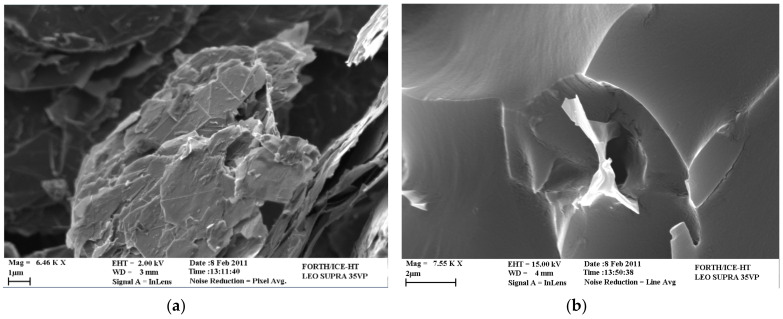
SEM micrographs of the (**a**) GNP agglomeration, and (**b**) buckled and rolled up GNP (0.5 wt.% Type 2 GNP nano-reinforced polymer).

**Table 1 materials-14-05828-t001:** Comparative results for the tested materials.

Properties	% Increase or Decrease Compared to the Neat Epoxy				
	GNPs Content wt.%				
	0.5 wt.%	1 wt.%	5 wt.%	10 wt.%	15 wt.%
Tg Type 1	−5%	−1.5%	−3%	−2%	−1.5%
Tg Type 2	−7%	−4%	−8%	−6%	−13%
Impact Resistance Type 1	+50%	+40%	+26%	−36%	−45%
Impact Resistance Type 2	+38%	+33%	−3%	−39%	−43%
Thermal Conductivity Type 1	0%	0%	+26%	+81%	+130%
Thermal Conductivity Type 2	0%	−4%	0%	+7%	+33%

## Data Availability

The data presented in this study are contained within the article.

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
