# Peer review of "On the Multi-Functional Behavior of Graphene-Based Nano-Reinforced Polymers"

_materials, 2021, doi:10.3390/ma14195828_

Round 1

Reviewer 1 Report

Dear Authors

The manuscript describes the assessment of the polymers performance reinforced by layered Graphene Nano-Platelets (GNPs). The manuscript presented concerns an interesting and actual subject. This manuscript can be accepted after major revision.

Some comments are listed below:

  1. The overall English needs to be improved. Please seek guidance from a native English speaker if possible (commas, "the" "a", and others should be corrected).
  2. Please add in the introduction some information about GNPs. Please cite (1) Appl. Sci. 2020, 10, 1753; doi:10.3390/app10051753  (2) Materials 2020, 13(21), 4975; https://doi.org/10.3390/ma13214975 (3)  Polymers 2020, 12(10), 2189; https://doi.org/10.3390/polym12102189
  3. Figure 1 please correct this image for better quality (the inscriptions on the drawing).
  4. Please correct  "Error! Reference source not found." lines: 138, 146, 154, 170, 225, 235, 240.
  5. Please add more sentences to the discussion of the characterization by SEM. Is EDS a good method to analyze elements such as C or O? Please explain.
  6. Please add TEM images and comments.
  7. Figure 6 and Figure 7 please correct these images for better quality (the inscriptions on the drawing). Please correct the scale in the figures
  8. Please add to the results part, the Raman spectra and ID/IG ratio and comments to the text about the quality of obtained materials?
  9. In conclusion, add some sentences about potential applications for enhancement.

Reviewer 2 Report

Dear authors, the formation of the polymer reinforced by Graphene is an exciting and important topic, which is proved with the number of articles. However, this is also the main problem: you need to clearly underline the novelty.

In general, the manuscript is readable, well arranged, and offers a compilation of nice results.

I believe in your results; nevertheless, the SEM analysis needs improvement as, in the present form, it is wishfull thinking only. It is necessary to perform at least EDX measurements.

Moreover, please improve the quality of all Figs.

I found some inconsistent (I guess) lines 122 – 126:

Let's assume that: “The graphene nanoplatelets (GNPs) consisted of 4 layers (Type 1) and the 12 layers (Type 2) were 97% pure… Type 1 has an average thickness of <4 nm, while Type 2 <12 nm. Both types have the same diameter of 2 microns…”, then how is it possible to get “the same surface area of 750 m2/g”? (By the way, what “the aspect ratio” is?)

I expect that the values are from the producer L

Finally, please write some conclusions, these are just the Summary.

Summing, I cannot find any drastic methodological mistakes (except SEM); in my opinion, this work has the potential to be published in Materials.

Reviewer 3 Report

The paper is well written but should be revised before acceptable for publication, few points are –

  1. Title of paper is questionable, authors state “multi-functional” in title but in paper they talk about effect of “number of graphene layers” and “aspect ratio” of GNPs?. Reviewer hope authors should match the title with scope of the work.
  2. The determination of number of layers for GNP flakes used in the work is questionable. Actually, for a GNP flake with thickness of <4nm should be having 12 layers and NOT 4 layers as described in the work. Actually, graphene is single atomic layer and for 1 nm thick flake, considering an interlayer distance of 0.34 nm between 2 adjacent graphene sheets, it should be 3 layers and NOT 1 layer as per author. To sum up, the interlayer distance between two graphene layers is 0.34 nm and NOT 1 nm. Please revise the number of layer in whole paper.
  3. Authors take the GNP flake thickness as per provided by the supplier. But it is NOT right, authors should experimentally prove this number. Most promising way is AFM microscopy or TEM microscopy? The number obtained from these tests should be reported and NOT what supplier claim?
  4. How aspect ratio is calculated and as per definition of aspect ratio, the aspect ratio calculation for GNP of type-2 is questionable. Moreover, the horizontal axis in GNP flake is NOT “diameter”, its “lateral dimension”. Moreover, the same surface area for two GNP flakes with different degree of exfoliation is questionable? It should be experimentally demonstrated via Adsorption isotherms?
  5. Introduction is well-written but it can be further improved by adding 3-4 references from Materials-MDPI journal on the subject of the work. Please discuss the advancement of the present work with the work published in same field in Materials.
  6. Many places in paper, its written “Error! Reference source not found”. Please cross check and correct all throughout the paper. What is the dimensions and ISO standards followed for determining mechanical, and thermal properties of the composites?
  7. Figure 5 is evident that the thermal conductivity strongly depends on number of layers and aspect ratio and increases significantly after 5 wt% of GNP? Please justify why?

Round 2

Reviewer 1 Report

Accept in present form

Author Response

Thank you for your review.

Reviewer 2 Report

Dear Authors, I still believe in you, but 4 questions and 0 answers:

1) did you know that you can not analyze SEM results as you did?  I have no idea what did you marked with arrows. More, (i) the presence of light and dark elements means better and worse conductive elements/places/areas; nothing more~ EDX or Micro-Raman or Micro~ FTIR should be performed to solve the problem (ii) while analyzing the composites (nanocomposites); why do your samples look like debris? Is there delamination occurs?

2) You must measure the SSA (low-temperature N2 adsorption isotherms) if you want to talk about the area; More: 97% most probably means here the elemental purity (97% of C in the sample); it does not mean that you have 97% of graphene, sorry. Raman analysis would be helpful here.

3) the quality of the figs is not better, sorry

4) the sentences you add as a concluding remark are (sorry, again) only wishful, having nothing in common with the results you presented.

Reviewer 3 Report

There are still many questionable areas in paper. It is not acceptable in current form. Major comments 

  1. The first point on title is not justified still. If authors want to talk about multi-functional behavior. They need to address chemistry part of the composites such as FTIR and XPS. They should talk about chemical functionality of two types of GNPs with polymer in nanocomposites or change the title.
  2. Please don't follow the supplier blindly and please follow scientific principles behind the aspects. Supplier say anything to sell their products.  To publish a scientific paper,  we have to justify the facts scientifically. As per supplier, only one thing is correct. either number of layers is correct or thickness of flake is correct. Bot are not correct at the same time scientifically. for example, a flake with 3 nm thickness and possess 3 number of layers is absolutely wrong scientifically. 
  3. Again, reviewer would be pleased of author justify number of layers or thickness of GNPs through some scientific technique and not rely on supplier. It is utmost required to publish this work. 
  4. The Figure legends are incorrect.  in some legends, authors use type-1 and type-2 GNP and other they mention 4 layers and 12 layers. Please crosscheck again. 
  5. In last paragraph of introduction, authors mention aspect ratio is crucial parameter in determining the properties while title says multi-functional aspects, this is not correct same time. 
